# Utilization of Multiparametric MRI of Prostate in Patients under Consideration for or Already in Active Surveillance: Correlation with Imaging Guided Target Biopsy

**DOI:** 10.3390/diagnostics10070441

**Published:** 2020-06-29

**Authors:** Jinxing Yu, Ann S. Fulcher, Sarah Winks, Mary A. Turner, William Behl, Anna Lee Ware, Nitai D. Mukhopadhyay, Candice Kim, Christopher Jackson, Harnek S. Bajaj, Lance J. Hampton

**Affiliations:** 1Department of Radiology, Virginia Commonwealth University Health System, Main Hospital, 3rd Floor, 401 North 12th Street, Richmond, VA 23298, USA; annfulcher@vcuhealth.org (A.S.F.); sarahwinks@vcuhealth.org (S.W.); mary.turner@vcuhealth.org (M.A.T.); 2School of Medicine, Virginia Commonwealth University Health System, Main Hospital, 401 North 12th Street, Richmond, VA 23298, USA; behlwc@vcu.edu (W.B.); kimc28@mymail.vcu.edu (C.K.); jacksoncr@mymail.vcu.edu (C.J.); bajajhs@mymail.vcu.edu (H.S.B.); 3Department of Biological Sciences, Virginia Polytechnic Institute and State University, Blacksburg, VA 24060, USA; annalee1@vt.edu; 4Department of Biostatistics, Virginia Commonwealth University Health System, 401 North 12th Street, Richmond, VA 23298, USA; nitai.mukhopadhyay@vcuhealth.org; 5Department of Urology, Virginia Commonwealth University Health System, 401 North 12th Street, Richmond, VA 23298, USA; lance.hampton@vcuhealth.org

**Keywords:** prostate cancer, active surveillance, multiparametric MRI, PI-RADS score, imaging-guided target prostate biopsy

## Abstract

This study sought to assess the value of multiparametric magnetic resonance image (mp-MRI) in patients with a prostate cancer (PCa) Gleason score of 6 or less under consideration for or already in active surveillance and to determine the rate of upgrading by target biopsy. Three hundred and fifty-four consecutive men with an initial transrectal ultrasound-guided (TRUS) biopsy-confirmed PCa Gleason score of 6 or less under clinical consideration for or already in active surveillance underwent mp-MRI and were retrospectively reviewed. One hundred and nineteen of 354 patients had cancer-suspicious regions (CSRs) at mp-MRI. Each CSR was assigned a Prostate Imaging Reporting and Data System (PI-RADS) score based on PI-RADS v2. One hundred and eight of 119 patients underwent confirmatory imaging-guided biopsy for CSRs. Pathology results including Gleason score (GS) and percentage of specimens positive for PCa were recorded. Associations between PI-RADS scores and findings at target biopsy were evaluated using logistic regression. At target biopsy, 81 of 108 patients had PCa (75%). Among them, 77 patients had upgrading (22%, 77 of 354 patients). One hundred and forty-six CSRs in 108 patients had PI-RADS 3 *n* = 28, 4 *n* = 66, and 5 *n* = 52. The upgraded rate for each category of CSR was for PI-RADS 3 (5 of 28, 18%), 4 (47 of 66, 71%) and 5 (49 of 52, 94%). Using logistic regression analysis, differences in PI-RADS scores from 3 to 5 are significantly associated with the probability of disease upgrade (20%, 73%, and 96% for PI-RADS score of 3, 4, and 5, respectively). Adding mp-MRI to patients under consideration for or already in active surveillance helps to identify undiagnosed PCa of a higher GS or higher volume resulting in upgrading in 22%.

## 1. Introduction

The aim of active surveillance (AS) is to avoid radical treatment and its side-effects in men who have truly low risk prostate cancer (PCa), whilst offering radical treatment to those men who are at higher risk of local progression or metastatic disease [1]. The traditional tools used to attribute these risk categories are prostate-specific antigen (PSA), digital rectal examination, transrectal ultrasound-guided biopsy (TRUS), and their repeated application over time [2]. However, none of these tools are sufficiently sensitive or specific to stratify patients. Despite careful clinical patient selection, up to 35% of men in AS will experience biopsy reclassification during follow-up and the resulting delays in treatment may negatively affect outcomes [3,4,5,6]. Most reclassifications can be explained by the under-sampling of more aggressive tumors at the time of initial TRUS biopsy rather than progression of indolent tumors [7,8]. The barrier to acceptance of AS for men with PCa is the risk of underestimating the cancer burden upon initial biopsy. Therefore, it is important to identify clinically significant PCa in patients who are already in or under consideration for the program of active surveillance.

Recently, mp-MRI has become a recognized technique with the highest accuracy for identifying and characterizing PCa and for differentiating these tumors from less aggressive tumor foci (i.e., those with a Gleason score ≤ 6) (GS) [9,10,11,12]. Imaging-guided target needle biopsy of visible cancer-suspicious regions (CSRs) at mp-MRI has also been proven to be an effective diagnostic strategy to better sample clinically significant and aggressive tumors and to avoid over-diagnosis of small- and low-risk tumors [13,14,15]. The result is that mp-MRI of prostate followed by imaging-guided target biopsy for the CSR is rapidly gaining acceptance as a means for evaluating patients with elevated PSA and prior negative TRUS biopsy. However, the utilization of prostate mp-MRI and imaging-guided biopsy for men who had a prior TRUS biopsy proven GS 6 PCa and under consideration for or already in AS has not been fully described. Further, mp-MRI currently is not included in the decision-making algorithms or criteria for AS. In the present study, our goals are to assess the value of mp-MRI in patients with PCa GS ≤ 6 and under consideration for or already in AS and to determine the rate of upgrading by confirmatory imaging-guided target biopsy.

## 2. Methods

### 2.1. Patients

From January 2012 to September 2017, 354 patients with low-grade PCa (GS ≤ 6) diagnosed by TRUS biopsy were referred to the department of radiology for mp-MRI studies (median interval from the TRUS bx to prostate MRI is 9.6 months). These patients were retrospectively identified in our database. Of the 354 patients, 241 were under consideration for AS and the remaining 113 had already enrolled in the program of AS. In order to be considered suitable candidates for AS, patients had to have low-grade, low-risk, localized PCa (clinical stage T1c-T2a, Gleason score 6 or less, no Gleason pattern 4 or 5, and serum PSA 10 ng/mL or less). No more than 3 cores on the entry biopsy could be positive for PCa and all biopsy cores had to have less than 50% involvement. Eligible candidates had to be between ages 50 and 80 years and have a life expectancy of more than 5 years [16,17]. The institutional review board approved this study (HM20005006, 7 September 2015).

All patients with CSRs at mp-MRI but without imaging-guided target biopsy were excluded (*n* = 11). Patients with prior diagnosis of prostatic intraepithelial neoplasia or atypia only were also excluded for the study (*n* = 35).

The patient’s age, PSA level, prostate volume, and PSA density were noted for all patients.

### 2.2. MR Imaging Acquisition

To identify suspicious lesions and plan for imaging-guided biopsy, all patients in the study group underwent mp-MRI (1.5 Tesla, Signa HDxt, GE Healthcare, Waukesha, WI, USA) with a body coil and an endorectal coil (Medrad, Warrendale, PA, USA) in place for signal reception. Axial, sagittal, and coronal T2-weighted images, diffusion-weighted images (DWI), and dynamic contrast-enhanced (DCE) MR images with IV contrast of MultiHance (in a dose of 0.2 mL/kg) were acquired.

All prostate MRI images were reviewed on a PACS workstation and an analytical software work station (Invivo, Gainesville, FL, USA) that projected the calculated DCE-MRI parameters as color overlays on T1-weighted images.

### 2.3. MR Imaging Interpretation

All prostate MRIs were interpreted by one of two experienced genitourinary radiologists immediately after completion of them. Cancer-suspicious regions were defined according to the previously published criteria for malignancy [18,19,20,21]. For both the peripheral and transitional zones, the reader identified and assigned PI-RADS scores for the likelihood of cancer on a five-point scale based on the PI-RADS v2 (1 = very low; 2 = low; 3 = indeterminate; 4 = high; 5 = very high). Therefore, the possibility for PCa would be low if the score was ≤2, moderate if the score was 3, and high if the score was ≥4. The PI-RADS score of each CSR was identified and recorded from the original prostate MRI reports by Jinxing Yu.

The evaluation was performed by using a combination of T2-weighted images, DWI, and an apparent diffusion coefficient map (ADC), and DCE images. The CSRs in the peripheral zone (PZ) were noted to demonstrate the following: (a) focal low signal intensity on T2-weighted images, (b) focal diffusion restriction, and/or (c) rapid contrast wash in and out. Diffusion restriction had to be present for the lesion in the PZ to be characterized as a CSR [19,20,21]. For the transitional zone (TZ), the CSRs were noted to demonstrate the following: (a) on T2-weighted image, homogeneous low signal intensity, ill-defined margins, lenticular shape, lack of capsule, and invasion of the anterior fibromuscular stroma, (b) focal diffusion restriction with ill-defined margins, and/or (c) rapid contrast wash in and out, that was not symmetric compared with the contralateral side of the prostate. The presence of abnormal findings on the T2-weighted images and diffusion weighted images was necessary for a lesion to be characterized as a CSR in the TZ [21]. If there were two or more suspicious areas in a given patient, the locations of these areas were recorded separately.

All MRI studies without PI-RADS scores based on PI-RADS v2 in their original reports (*n* = 55) were retrospectively reviewed by one reader who then assigned a PI-RADS score for each CRS. The reader was aware that the patients might have undergone imaging-guided prostate biopsy but was blinded to the biopsy results.

Finally, MR imaging findings and PI-RADS scores were directly compared with the pathological results of the imaging-guided target prostate biopsy by an experienced GU radiologist (Jinxing Yu).

### 2.4. Imaging Guided Target Prostate Biopsies

Imaging-guided target prostate biopsy included in-bore MRI-guided prostate biopsy (MRGB) (January 2012 to August 2014, *n* = 186) and ultrasound/MRI (US/MRI) fusion-guided prostate biopsy (September 2014 to September 2018, *n* = 446) (median interval from the prostate MRI to imaging-guided bx is 1.6 months). Before biopsy, the patients were given a cleansing Fleet^®^ enema and antibiotic prophylaxis per the American Urological Association guidelines. All patients underwent monitored conscious sedation for the procedure.

MRGB was performed on a 1.5 Tesla scanner (Siemens Magneton Avanto, Erlanger, Germany). Patients were placed in a prone position, then a needle guide filled with a Gd-chelate dotted gel for visualization and fixed to a portable biopsy device (Invivo, Gainesville, FL, USA) was introduced rectally. Axial T2WI and/or DWI were acquired as baseline images for targeting. Biopsies were obtained from CSRs with an MRI compatible 18-gauge core needle biopsy gun (Invivo). All biopsies were performed by a single radiologist. At least two specimens were taken from each CSR. The median time of MRGB was 43 min (Range: 35–85 min).

US/MRI fusion-guided prostate biopsies were performed transrectally with patients in the left decubitus position. The mp-MRI was interpreted by the radiologists; the images were segmented, and lesion locations were recorded (Dynacad, Invivo, Amsterdam, The Netherlands). Patients with lesions identified on mp-MRI underwent a target biopsy performed by either a radiologist or urologist. First, a 2-dimensional TRUS sweep of the prostate was performed in the axial plane to render a 3-dimensional ultrasound image that was registered and fused to the pre-biopsy MRI using a UroNav device (Invivo, Philip, Amsterdam, The Netherlands). Then, the lesions identified at mp-MRI were superimposed using the T2-weighted sequence on the real-time TRUS images. Each lesion was sampled both in the axial and sagittal planes by an end-fire TRUS probe (Philip, Amsterdam, The Netherlands) with at least 2 biopsy cores taken.

### 2.5. Statistical Analysis

We used contingency tables to assess the accuracy of mp-MRI to predict cancer upgrading for patients deemed ineligible for AS based on confirmatory imaging-guided target biopsy. Associations between PI-RADS scores and findings at imaging-guided target biopsy were evaluated using logistic regression. The Mann–Whitney *U* test was performed to compare mean age, PSA, prostate volume, and PSA density between the upgrading and non-upgrading patients’ groups. A significance level of <0.05 was used for all analyses.

## 3. Results

Overall, 354 patients were included in this study. Clinical characteristics and prior TRUS biopsy of the patients are shown in Table 1.

As noted in Figure 1, the mp-MRI of the cohort (*n* = 354) did not detect CSRs in 198 patients (56%), while MRI and initial biopsy were concordant in 37 (10%). MRI detected CSRs in 119 patients (34%). Imaging-guided target biopsies were performed in 108 of 119 patients for a total of 146 CSRs. Eleven of 119 patients did not have an imaging-guided biopsy, of whom 7 refused, 2 died from other illnesses, and 2 were lost to follow-up.

At confirmatory imaging-guided target biopsy, 81 of 108 patients had biopsy-proven PCa (75%). Among them, 77 patients had upgrading (22%, 77 of 354 patients) (GS 7 *n* = 50, GS 8 *n* = 14, GS 9 *n* = 4, and GS 6 *n* = 9 with the biopsy specimen positive for PCa ≥ 50%). The remaining four patients had PCa GS 6 < 50% which did not result in upgrading. All 77 patients underwent definite treatment (surgery *n* = 41 and radiation *n* = 36). In the study, a total of 29% of patients (31/108) who underwent imaging-guided target biopsy revealed no or low-risk PCa

The PI-RADS scores for 146 CSRs in the 108 patients were PI-RADS 3 (*n* = 28), 4 (*n* = 66), and 5 (*n* = 52). The upgraded rate for each category of CSR was for PI-RADS 3 (5 of 28, 18%), 4 (47 of 66, 71%), and 5 (49 of 52, 94%) (Table 2).

In the 77 patients with upgrading, 41 patients (53%) had dominant tumors detected in the transitional zone (Figure 2) and the remaining 36 patients (47%) had dominant tumors detected in the peripheral zone. The most common site of the tumor was in the anterior aspect of the transitional zone (35 of 77, 45%) (Figure 3), followed by the PZ at the apex in 21 (27%) (Figure 4), PZ paramedian region in 5 (7%), PZ anterior horns in 4 (5%), and other regions in 12 (16%).

Of 198 patients with initial negative mp-MRI studies and 37 with initial concordant findings, 48 of them had follow-up mp-MRIs (35 with one, 9 with two, and 4 with 3 follow-ups). Forty-two of 48 patients were again negative on follow-up MRIs. Three had an interval increase in the size of the initial concordant lesions and three had new CSRs away from the initial positive biopsy site. On further imaging-guided target biopsy, three had upgrading and the remaining three did not.

Using logistic regression analysis, the predicted upgrading probability is 22%, 73%, and 96% for PI-RADS score 3, 4 and 5, respectively. The P value associated with PI-RADS score is 0.000539 which is statistically significant. As a logistic regression model uses a linear relationship between the log odds and the score, in the range of PI-RADS score 3 to 5, each unit increase in the PI-RADS score is predictive of a 2.25 unit increase in the log odds of upgrading (Table 3).

In the Mann–Whitney *U* test, the mean age, PSA, and prostate volume between those upgrading (*n* = 77 patients) compared with those non-upgrading (*n* = 266 patients) are not statistically significant except for PSA density (median 0.18 vs. 0.11 ng/mL/cc, *p* = 0.0008).

## 4. Discussion

The use of mp-MRI for the diagnosis and characterization of PCa has increased significantly over the past six years. Using radical prostatectomy specimens as a reference, an excellent performance of mp-MRI was demonstrated in identifying tumors of the peripheral and transition zones [22,23]. The application of this technology has been extended to many clinical indications such as identifying tumors in men with previous benign biopsies and persistently elevated PSA levels [24,25]. Despite the increasing use of MRI in PCa diagnosis and management, the use of such technology in AS populations remains undefined and underused. Our research results indicate that mp-MRI may have a very important role not only in screening patients for entry into AS but also in reaffirming patients' eligibility for continuing on AS.

An important finding of our study is that mp-MRI revealed unrecognized lesions with imaging-guided biopsy confirming that the lesion was clinically significant and resulted in upgrading in up to 22% of patients (no longer fulfilling criteria for AS). Further, when CSRs were identified on mp-MRI with PI-RADS 4 and 5, the rate for upgrading was 71% and 94%. To our knowledge, this is one of a few studies in which investigators assigned PI-RADS score to each CSR based on PI-RADS v2 and then used imaging-guided biopsy as a reference test to confirm the CSRs identified on prior mp-MRI in AS patients.

Imaging-guided target biopsy of the prostate such as ultrasound/MRI fusion biopsy has emerged as an important tool in the confirmation of CSRs identified on the mp-MRI [26,27]. Compared with the standard TRUS-guided biopsy scheme, MR-US fusion biopsy is associated with increased detection of clinically relevant cancer and decreased detection of low-risk cancers [26,27,28]. The fusion method permits the targeting of biopsies into regions of interest identified on mp-MRI, resulting in a higher tumor detection yield (Figure 1 and Figure 2). TRUS-guided prostate biopsy serving as confirmatory biopsy might miss sampling the CRSs again because these CRSs are commonly located in the regions anatomically difficult to reach by TRUS biopsy, such as in the anterior or inferior aspects of the prostate. That is why it is common to detect significant PCa by mp-MRI in patients who might have had multiple prior TRUS biopsies [25,26,27]. Since imaging-guided prostate biopsy has been available for less than 10 years and in a few large teaching hospitals, several prior research studies assessing the utilization of mp-MRI in patients on AS used TRUS biopsy as their confirmatory tools for CSRs on mp-MRI [29,30], which limits the ability to compare our results with their findings. Ideally, a confirmatory repeat biopsy for any CSR on mp-MRI should be carried out by MRI or MRI/TRUS fusion image-guided target biopsy [27]. The combination of imaging-guided target biopsy and systematic TRUS biopsy may offer additional benefit particularly for those patients who did not have a prior TRUS biopsy because it can detect clinically significant tumors that are occult on mp-MRI even on retrospective evaluation due to a sparse growth pattern and a low malignant epithelium–stroma ratio [31,32].

Although our study was not designed to validate the PI-RADS v2 scoring system, our results do show that PI-RADS v2 is a promising scoring system for the differentiation between patients suitable for AS and patients needing upgrading, the latter requiring radical treatment. In our study, patients with PI-RADS 4 and 5 had a rate for upgrading in 71% and 94%, respectively. The result was comparable with those of Hoeks et al., who reported a sensitivity of 92% for the detection of PCa ≥ 7 in the case of higher PI-RADS scores (≥4) [33]. Vargas et al. reported a sensitivity of 87% to 96% for biopsy upgrading in the case of a predefined MR imaging score of 5 for cancer presence [34]. This finding may indicate that a patient with CRS at mp-MRI of PI-RADS 4 or 5 is likely no longer fulfilling the criteria of AS.

Our study population included patients not only under consideration for AS but also already in AS if these patients had not had prior mp-MRIs and did not have a confirmatory TRUS biopsy. We designed our research in this way because these patients had the risk that prior biopsy might have under sampled or completely missed their dominant lesions. Given that many of the missed dominant tumors were located in the anterior aspect of the prostate (45%) or in the apex (27%), a blind, repeat TRUS biopsy would not likely identify these tumors. In contrast, mp-MRI is very accurate in detecting these missed significant tumors. In addition, combining mp-MRI with patients’ PSA density information may further help identify patients who harbor significant and high-grade PCa as our study demonstrated PSAD was higher in patients with upgrading compared with those with non-upgrading [35].

Study limitations: First, there is limited follow-up for our patients with normal MRIs. Our mp-MRI studies and imaging-guided target biopsies were performed in a tertiary referral center where AS patients were referred to us from many centers in the state and the country. After receiving normal results of prostate MRI from our center, patients often went back to their local urologists for AS, which created difficulty in following-up for these patients. Nonetheless, the study included 46 patients with follow-up MRIs: 3 of them had upgrading. Margel et al. reported 3.5% reclassification during AS follow-up when patients had initial normal MRIs [29]. In addition, our results could provide the foundation for a prospective study incorporating all follow-up and outcome data. Second, our study included patients who had MRI-guided or US/MRI fusion-guided target biopsies performed during a period of about six years. This allowed us to maximize the number of eligible patients but also introduced potential diagnostic heterogeneity. However, we found no significant differences in tumor detection rate of MRI CSRs between the MRI-guided and US/MRI-guided target biopsies. Instead, the outcome of imaging-guided target biopsy is directly related to the PI-RADS scores of CSRs [33].

In conclusion, adding mp-MRI of the prostate for patients under consideration for or already in active surveillance helps to identify undiagnosed clinically significant PCa results in upgrading in 22% of these patients, and reaffirms others’ eligibility for active surveillance. The rates of upgrading for patients with PI-RADS 4 and 5 cancer suspicious regions at mp-MRI were 71% and 94%, respectively. The result indicates that the standardized reported mp-MRI using PI-RADS v2 may be a promising tool for the selection of patients suitable for AS.

## Figures and Tables

**Figure 1 diagnostics-10-00441-f001:**
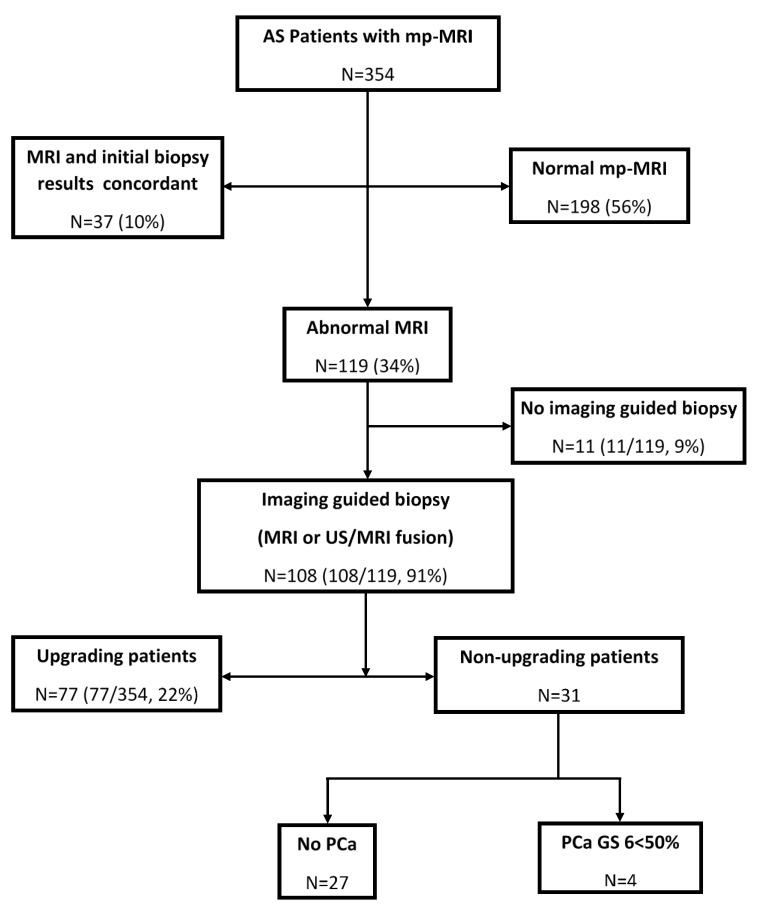
Flowchart showing patient selection.

**Figure 2 diagnostics-10-00441-f002:**
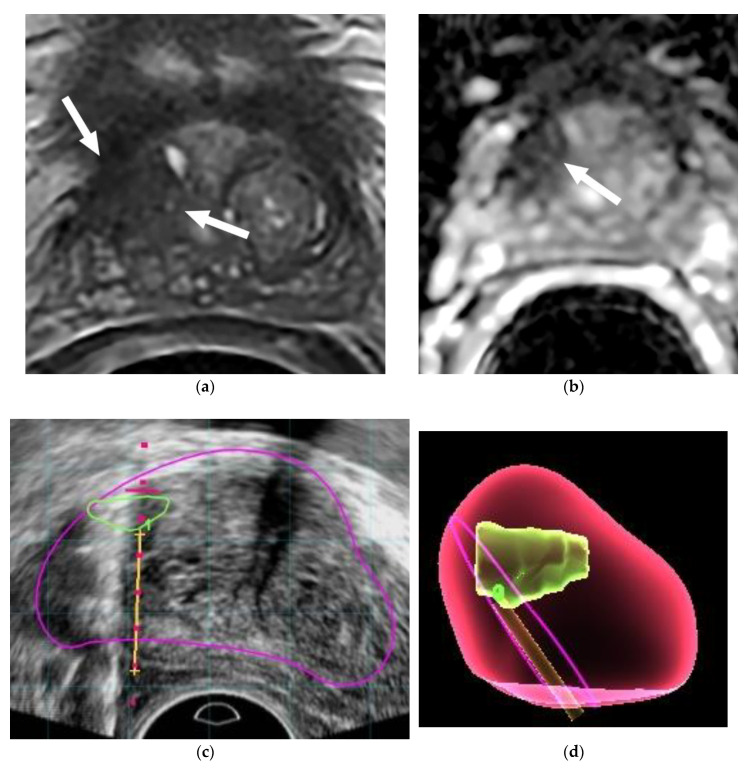
(**a**–**d**) MRI and ultrasound (US)/MRI fusion biopsy confirmed a large PCa GS 7 missed by the prior TRUS biopsy in a 68-year-old man with PSA 4.2 ng/mL. The patient is under consideration for active surveillance (AS). (**a**) Axial T2-weighted MR image demonstrates a low-T2 signal intensity lesion (arrows) in the lateral aspect of the right base transitional zone (TZ); (**b**) apparent diffusion coefficient (ADC) image demonstrates the lesion (arrow) seen at T2WI with diffusion restriction in the lateral aspect of the right base TZ (PI-RADS 5); (**c**) a real-time axial transrectal ultrasound is performed to assist with needle guidance at the time of MR/ultrasound fusion biopsy. The MR/ultrasound fusion platform overlays the outline of the lesion suspicious for prostate cancer (green line) and contour of the prostate (pink line). A dotted red line demonstrates the path of the needle, and when a biopsy is performed, the location can be recorded as shown here with the yellow line; (**d**) a 3-dimensional map from the data above is generated at the conclusion of the biopsy, demonstrating the contour of the prostate (red), the location of the tumor lesion (green), and the location of the targeted MR/ultrasound fusion biopsies (yellow and pink lines).

**Figure 3 diagnostics-10-00441-f003:**
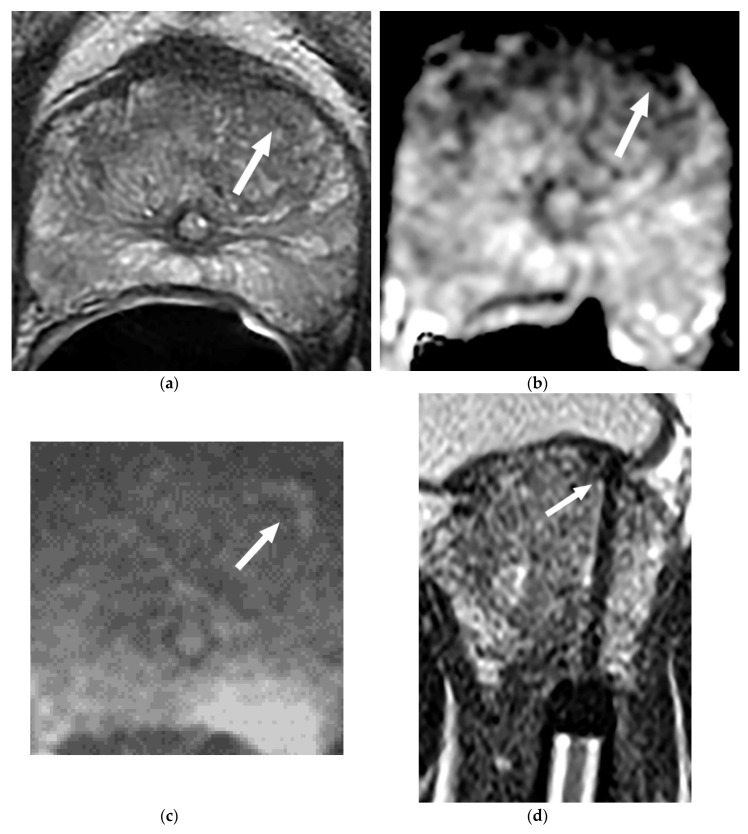
(**a**–**d**) MRI and MRI-guided biopsy confirmed a small PCa GS 7 missed by the prior TRUS biopsy in a 64-year-old man with PSA 7.4 ng/mL. The patient has been on AS for one year. (**a**) Axial T2-weighted MR image demonstrates a small low-T2 signal intensity lesion (arrow) in the anterior aspect of the left mid TZ; (**b**) ADC image demonstrates the lesion (arrow) seen at T2WI with diffusion restriction in the anterior aspect of the left mid TZ (PI-RADS 4); (**c**) diffusion-weighted images (DWI) b = 1000 image demonstrates the lesion (arrow) seen in the anterior aspect of the left mid TZ with a bright signal intensity consistent with diffusion restriction; (**d**) Axial T2-weighted MR image during MRI-guided prostate biopsy (MRGB) session demonstrates the biopsy needle through the lesion (arrow) at the anterior aspect of the left mid TZ.

**Figure 4 diagnostics-10-00441-f004:**
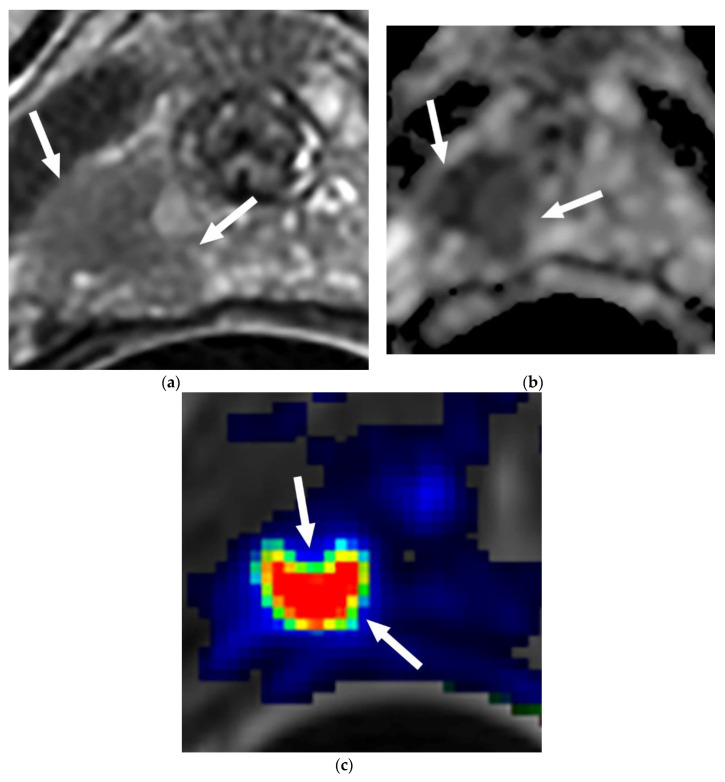
(**a**–**c**) MRI and US/MRI fusion biopsy confirmed PCa GS 7 missed by the prior TRUS biopsy in a 65-year-old man with PSA 3.9 ng/mL. The patient is under consideration for AS. (**a**) Axial T2-weighted MR image demonstrates a low-T2 signal intensity lesion (arrow) in the right apex peripheral zone (PZ); (**b**) ADC image demonstrates the lesion (arrows) seen at T2WI with diffusion restriction in the right apex PZ (PI-RADS 5); (**c**) dynamic contrast-enhanced (DCE)-MRI color-coded map shows the lesion with high vascular permeability (arrows) in the right apex PZ.

**Table 1 diagnostics-10-00441-t001:** Clinical and transrectal ultrasound-guided (TRUS) biopsy characteristics of the 354 patients.

Variable	Value
Patients	354
Median age	Year; 63.4 (50–79)
Median PSA, ng/mL	6.1 (2.8–9.7)
Median prostate volume, cc	41 (18–159)
Median PSA Density, ng/mL/cc	0.15 (0.07–0.31)
Number of prior biopsy (%)	1 (198)
	2 (112)
	≥3 (44)
Number of positive biopsy core	1 (258, 73%)
	2 (96, 27%)
Biopsy Gleason score at TRUS biopsy	3 + 3 (354, 100%)

**Table 2 diagnostics-10-00441-t002:** Increased PI-RADS scores were associated with higher rates of upgrading (odds ratio 2.25–3.46, *p* < 0.001).

PI-RADS Score	Imaging-Guided bx of CSRs Upgrade		
	No	Yes	Total
3	23 (82%)	5 (18%)	28
4	19 (29%)	47 (71%)	66
5	3 (6%)	49 (94%)	52
Total	45	101	146

**Table 3 diagnostics-10-00441-t003:** Logistic regression of upgrading probability on PI-RADS score.

	Coefficient Estimate	Standard Error	*p* Value
Intercept	−8.004	2.54	0.0016
PI-RADS	2.245	0.64	0.00054

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
