# Peer review of "Utilization of Multiparametric MRI of Prostate in Patients under Consideration for or Already in Active Surveillance: Correlation with Imaging Guided Target Biopsy"

_diagnostics, 2020, doi:10.3390/diagnostics10070441_

Round 1
Reviewer 1 Report
I would like to have known the interval between initial biopsy and study mpMRI and biopsy. The authors showed understaging where the mpMRI lesion was anterior and this had not been sampled by the initial transrectal prostate biopsy.
I could not say whether the 22% upstaging was real or just inadequate initial staging due to the known inaccuracy of transrectal biopsy.
The study demonstrated the utility of fusion biopsies.
At least in the UK every man has a mpMRI before their biopsy, which is increasingly a transperineal biopsy. In the USA this would appear not to be the case, at least in the study period.
I agree that standardised reporting of MRI using PIRAD v2 scores should be adopted.
Author Response
Reviewer #1:
Comments and Suggestions for Authors
- I would like to have known the interval between initial biopsy and study mpMRI and biopsy. The authors showed understaging where the mpMRI lesion was anterior and this had not been sampled by the initial transrectal prostate biopsy.
Done. The sentence “(Median interval from the TRUS bx to prostate MRI is 9.6 months)” was added in Line 71 to 72.
The sentence “(Median interval from the prostate MRI to imaging guided bx is 1.6 month)” was added in Line 123.
- I could not say whether the 22% upstaging was real or just inadequate initial staging due to the known inaccuracy of transrectal biopsy.
We truly believe that 22% upgrading was directly related to the initial inaccurate sampling by TRUS biopsy and these missed lesions were mainly located either far anterior or inferior.
No change made.
- The study demonstrated the utility of fusion biopsies.
Agree!
- At least in the UK every man has a mpMRI before their biopsy, which is increasingly a transperineal biopsy. In the USA this would appear not to be the case, at least in the study period.
During the studying period in the US, almost every man had TRUS biopsy before they had prostate mp-MRI. Now in our institute, about 30% patients had prostate mp-MRI without a prior TRUS biopsy.
- I agree that standardised reporting of MRI using PIRAD v2 scores should be adopted.
Agree!
Reviewer 2 Report
It is a very carefully designed and reported work on application of multiparametric MRI followed by image-guided biopsy in patients with prostate cancer risk.
The conclusions, showing significant improvement in detection of otherwise difficult to identify lesions are well justified by the obtained results.
I have no big critical issue to report.
Just few minor points:
L91
Is it T1 or T2 ? T2 weighted image is earlier mentioned. Unless here T1 is directly related to DCE.
L100 & L118 Information in brackets is missed
L162 change "core" to "score"
Author Response
Reviewer #2:
Comments and Suggestions for Authors
- It is a very carefully designed and reported work on application of multiparametric MRI followed by image-guided biopsy in patients with prostate cancer risk. The conclusions, showing significant improvement in detection of otherwise difficult to identify lesions are well justified by the obtained results. I have no big critical issue to report.
Thanks!
- Just few minor points:
L91, Is it T1 or T2 ? T2 weighted image is earlier mentioned. Unless here T1 is directly related to DCE.
Done. L91, The sentence “……projected calculated DCE-MRI parameters as color overlays on T1-weighted images.” Here it is T1WI directly related to DCE.
No change made.
L100 & L118 Information in brackets is missed
L100 and L118: The sentence “.... original prostate MRI reports by ( ).” and the sentence “….by an experienced GU radiologist ( ).”
Done. (JY) was added in Line 101 and L119.
L162 change "core" to "score"
L162: The sentence “(……..and GS 6 n=9 with positive core ≥50%)." in L164
Reviewer 3 Report
Please see attached word document for corrections, questions and comments on this paper. All authors should review the manuscript for English and content. Was there an age related correlation with upstaging or upgrading? The study design would be bias to this but possibly PCa in elderly men are more stable and less liable to progress. I am concerned about the numbers after the reported exclusion at the begining of the paper.

Author Response
Reviewer #3:
Comments and Suggestions for Authors
Line 20- “To assess value of multiparametric MRI (mp-MRI) in patients….” Suggest ‘This study sought to assess…’
Done. “To assess value of multiparametric MRI (mp-MRI) in patients….” changed to ‘This study sought to assess…’ in Line 20.
Line 22- “354 consecutive men….” Should not start a sentence with a digit, it should be spelled out. Suggest ‘Three hundred and fifty-four consecutive men…’ NB there are multiple instances of this and it makes the article difficult to read with the references.
Done. “354 consecutive men….” changed to ‘Three hundred and fifty-four consecutive men…’ in Line 22.
Similar changes have also been made in Line 25 (119, One hundred and nineteen), in Line 26 (108, One hundred and eight), and in Line 31 (146, One hundred and forty-six).
Line 45- “However, none of these tools is sufficiently sensitive or specific…” Suggest ‘However, none of these tools are sufficiently sensitive or specific…’
Done. “However, none of these tools is sufficiently sensitive or specific…” changed to ‘However, none of these tools are sufficiently sensitive or specific…’ in Line 46.
Line 54- “Recently, multiparametric (mp) magnetic resonance imaging (MRI) has become…” Abbreviations have already been introduced no need to repeat.
Done. “Recently, multiparametric (mp) magnetic resonance imaging (MRI) has become…” change to “Recently, mp-MRI has become…”in Line 55.
Line 66- “…consideration for or already in AS and and to determine the rate of…” Remove repeated and.
Done. “..and..” deleted in Line 67.
Line 71- “TRUS biopsy were referred to our department for mp-MRI studies…” Whose department radiology or urology?
Done. The sentence “….TRUS biopsy were referred to our department for mp-MRI studies…” changed to “….TRUS biopsy were referred to the department of Radiology for mp-MRI studies…” in Line 71.
Line 72- “…72 retrospectively identified in our database.” If they were “referred” how were they already in your database? Whose database and how is it managed?
Done. The related sentences “From January 2012 to September 2017, 354 patients with low grade PCa (GS ≤ 6) diagnosed by TRUS biopsy were referred to our department for mp-MRI studies. These patients were retrospectively identified in our database.” In our department, all patients with prostate mp-MRI were recorded in our PACS and kept in a database managed by a prostate nurse who also helped schedule all prostate biopsies.
No change made.
Line 79- “All patients with CSRs at mp-MRI but without imaging guided target biopsy were excluded (n=11).” Doesn’t fit with description of patient referral in line 70.
Done. L70 sentence “Of the 354 patients, 241 were under consideration for AS and the remaining 113 had already enrolled in the program of AS.” The sentence indicated that these 354 patients were either on AS or under consideration for AS. The 11 cases met the AS criteria clinically, so they were included here.
L79: “All patients with CSRs at mp-MRI but without imaging guided target biopsy were excluded (n=11).” I had to exclude them (n=11) because they did not come back for a target biopsy. I did not know if these CSRs on mp-MRI in these 11 patients would be positive or negative for PCa by target biopsy.
No change was made.
Line 80- “Patients with prior diagnosis of prostatic intraepithelial neoplasia or atypia were also excluded for the study (n=35).” Were these patients with PIN and Atypia only?
Yes. The sentence “Patients with prior diagnosis of prostatic intraepithelial neoplasia or atypia were also excluded for the study (n=35).” changed to “Patients with prior diagnosis of prostatic intraepithelial neoplasia or atypia only were also excluded for the study (n=35).”in Line 81-82.
Line 100- “prostate MRI reports by ( ). Something is missing here?
Done. JY was added to here in Line 101.
Line 103- “…were noted to demonstrate a) focal low signal intensity on T2-weighted images, b)…” Suggest ‘…were noted to demonstrate; a) focal low signal intensity on T2-weighted images, b)…’
Done. “…were noted to demonstrate a) focal low signal intensity on T2-weighted images, b)…” changed to ‘…were noted to demonstrate; a) focal low signal intensity on T2-weighted images, b)…’. in Line 104.
Line 106- “…noted to demonstrate a) on T2-weighted image…” Same thing, demonstrate;
Done. “…noted to demonstrate a) on T2-weighted image…” changed to “…noted to demonstrate; a) on T2-weighted image…”in Line 107.
Line 113- “All MRI studies without PI-RADS scores based on PI-RADS v2 in their original reports (n=55) were retrospectively reviewed by one reader who then assigned PI-RADS score for each CRS.” This needs to be explained further. Line 70 states that patients were selected based on TRUS PBx results. Was the radiology department using an earlier PIRADS version that had to be converted during the time frame of this study?
Yes. PI-RADS v2 was available about at the end of 2014. So the cases (n=55) had been assigned PI-RADS scores using new version.
No change was made.
Line 118- “…pathological results of the imaging guided target prostate biopsy by an experienced GU radiologist ( ).” Missing something? Was the pathology reviewed by a GU pathologist or central pathologist?
Done. JY was added to ( ) in Line 119.
Yes. The pathology in all patients were reviewed by experienced GU pathologists.
Line 129- “…CSRs with an MRI compatible 18-gause core needle biopsy gun (Invivo).” Suggest ‘…CSRs with an MRI compatible 18-gauge core needle biopsy gun (Invivo).’
Done. “…CSRs with an MRI compatible 18-gause core needle biopsy gun (Invivo).” changed to ‘…CSRs with an MRI compatible 18-gauge core needle biopsy gun (Invivo).’in Line 131.
Line 135- “…underwent a target biopsy performed by one physician (radiologist or urologist).” Suggest ‘…underwent a targeted biopsy performed by either a radiologist or urologist.’
Done. “…underwent a target biopsy performed by one physician (radiologist or urologist).” changed to ‘…underwent a targeted biopsy performed by either a radiologist or urologist.’ In Line 137.
Line 150- “Overall, 354 patients were included in this study.” Should this not be ‘Overall, 308 patients were included in this study.’ [46 excluded.]
Done. During that time period, there were 354 patients who were in AS or under consideration for AS, so they were included in the study, including these 11 cases. Please also see the response to L79.
35 cases with prior diagnosis of prostatic intraepithelial neoplasia or atypia only by TRUS biopsy were excluded for the study (not part of 354) because they were not in AS. We like to emphasize this here that these cases did not meet criteria for AS in our institute.
No change made.
Line 152- Table 1. Did PSA density correlate with increasing PIRADS score?
Done. The purpose of the study was to assess value of mp-MRI in patients in or under consideration for AS. The correlation between the PSA density and PI-RADS score was not studied, partially due to a relatively small sample size (108 patients with target biopsies).
No change made.
Line 155- “Imaging guided target biopsies were performed in 108 of 119 patients for a total of 146 CSRs.” Should the total not be 108 + 37 = 145?
Done. In 119 patients who had CSRs, only 108 patients had target biopsies. Some of them (n=108) might have more than one CSRs at mp-MRI, so the total CSRs were 146.
No change made.
Line 159- Figure 1. The numbers don’t add up correctly if 46 patients were excluded.
Done. Please see the response to Line 79 and L150.
No change was made.
Line 206- “P value associated with PI-RADS score is 0.000539 which is statistically significant.” This presentation of P value needs to be reviewed by a statistician.
Yes. All statistical analysis was made by a statistician.
Line 210- “…(n=266 patients) are not statistically significant except for PSA density (median 0.18 vs 0.11 ng/ml/cc, p=0.0008).” Why not use a correlation coefficient?
Done. We have been undergoing a project which will include more than 500 target biopsy patients. In that study, correlation between the PSA density in patients with and without PCa, and in patients with different PI-RADS scores will be carefully studied.
No change made.
Line 223- “…guided biopsy confirming that the lesion was clinically significant and resulting in upgrading in up…” Suggest ‘…guided biopsy confirming that the lesion was clinically significant and resulted in upgrading in up…’
Done. “…guided biopsy confirming that the lesion was clinically significant and resulting in upgrading in up…” changed to ‘…guided biopsy confirming that the lesion was clinically significant and resulted in upgrading in up…’ in Line 229.
Line 224- “…to 22 % of patients (no longer fulfilling criteria for AS).” How were these patients managed? Did they choose to come off of surveillance?
Done. The sentence “All 77 patients underwent definite treatment (surgery n=41 and radiation n= 36).” was added to the Results, in Line 165-166.
Line 232- “…increased detection of clinically relevant cancer and decreased detection of low-risk cancers…” In this current study still 29% of the patients underwent image guided PBx revealing no or low risk PCa. The author should state this rate.
Done. The sentence “In the study, a total of 29% of patients (31/108) who underwent imaging guided target biopsy reveal no or low risk PCa.” was added to the Results, in Line 166 to 167.
Line 238- “…might have multiple prior TRUS biopsies…” suggest ‘…might have had multiple prior TRUS biopsies.’
Done. “…might have multiple prior TRUS biopsies…” changed to ‘…might have had multiple prior TRUS biopsies.’ In Line 244.
Line 238- “Since imaging guided prostate biopsy has been available approximately 6 years and in a few large teaching hospitals….” Image guided PBx has been available for more than 6 years. Is this specific to their institution?
Done. US/MRI fusion biopsy has been available in the United States since 2014, about 6 years and MRI guided biopsy in 2011, about 9 years. So the sentence “Since imaging guided prostate biopsy has been available approximately 6 years and in a few large teaching hospitals….” changed to “ Since imaging guided prostate biopsy has been available in less then 10 years and in a few large teaching hospitals…..” in Line 245.
Line 241- “…which resulted in the difficulty to compare our results with their findings.” Suggest ‘…which limits the ability to compare our results with their findings.’
Done. “…which resulted in the difficulty to compare our results with their findings.” changed to ‘…which limits the ability to compare our results with their findings.’ In Line 247.
Line 242- “…carry out by using imaging guided target biopsy…” Suggest the authors are trying to say ‘…carry out by using MRI or MRI/TRUS fusion image guided target biopsy.’
Done. “…carry out by using imaging guided target biopsy…” changed to ‘…carry out by using MRI or MRI/TRUS fusion image guided target biopsy.’ In Line 248-249.
Line 243- “The combination of imaging guided target biopsy and systematic TRUS biopsy may offer additional benefit because it can detect some tumors that are occult on mp-MRI even on retrospective evaluation because of a sparse growth pattern and a low malignant epithelium stroma ratio [31,32].” The whole premise of using mp-MRI is to avoid detection of low risk PCa. Hence, this doesn’t fit in their discussion unless they want to discussion the false positives in their study as previously pointed out.
Done. We do agree that using mp-MRI can reduce the detection of low risk PCa. But we also know that mp-MRI may miss clinically significant PCa up to 10%. So adding the systematic TRUS biopsy to the target biopsy is important particularly for those patients who did not have prior TRUS bx.
The sentence “The combination of imaging guided target biopsy and systematic TRUS biopsy may offer additional benefit because it can detect some tumors that are occult on mp-MRI even on retrospective evaluation because of a sparse growth pattern and a low malignant epithelium stroma ratio [31,32].” changed to “The combination of imaging guided target biopsy and systematic TRUS biopsy may offer additional benefit particularly for those patients who did not have prior TRUS biopsy because it can detect clinically significant tumors that are occult on mp-MRI even on retrospective evaluation due to a sparse growth pattern and a low malignant epithelium stroma ratio [31,32].” In Line 250-252.
Line 282 “…for patients with PI-RADS 4 and 5 cancer suspicious regions at mp-MRI were 71% and 94%, respectively.” Again, what treatment did these patients receive? Did this intervention result in a change of management? The discussion would benefit from a discussion of added costs?
Done. Please see the response on Line 224.
The study of costs was complicated and was not the goal of our study.
Overall an interesting paper. Reasonably well written. Needs to have the English reviewed for English as a second language.
Thanks! Following changes based on an English reviewer.
Line 26: “to” changed to “a”
Line 28: “positive specimen” changed to “specimens positive for PCa”
Line 33-34: “changes” changed to “differences” and “is” changed to “are”
Lines 116: “….were…” changed to “was”
Line 240: “served as…” changed to “serving as…”
Line 242: rewritten as “…regions anatomically difficult to reach…”
Line 263: “…includes….” changed to “….included….”
Line 277: rewritten as “…following-up…”
Reviewer 4 Report
General Comments
The authors intended to figure out the positive rate of prostate target biopsy according to the PI-RADS score during or before active surveillance in 354 patients with low-risk prostate cancer of Gleason 3+3 in up to 2 cores. And the authors showed that the positive rate was increased according to PI-RADS score. Although the information of the manuscript is informative, the manuscript is not reached to the quality of publication because of unrelated Figures of the results and lack of important Table.
Major revisions
- The number of MRI-guided and ultrasound-guided biopsy should be shown in the manuscript.
- What are the purpose of showing Figure 3, 4, and 5? Are these representatives? There are no methods or results descriptions of the Figures in the result section of the manuscript according to the Figures. If it is representatives of some special methods or results, the authors should describe them according to the Figures. If it is just the representatives of the MRI-guided and ultrasound-guided biopsy, are these Figures necessary? Because there is no novel information in these Figures.
- There is no Table showing the results of logistic regression analysis. Please add and explain it.
Minor revisions
- In Figure 1, an abbreviation of bx should be spelled out.
- In Figure 1, the percent of 354 patients should be shown in N = 37, 198, 11, 31, 27, and 4 except for Upgrading patients of N = 77.
Author Response
Reviewer 4:
General Comments
The authors intended to figure out the positive rate of prostate target biopsy according to the PI-RADS score during or before active surveillance in 354 patients with low-risk prostate cancer of Gleason 3+3 in up to 2 cores. And the authors showed that the positive rate was increased according to PI-RADS score. Although the information of the manuscript is informative, the manuscript is not reached to the quality of publication because of unrelated Figures of the results and lack of important Table.
Thanks for your comments. A major revision was made which would improve the quality of the manuscript significantly.
Major revisions
- The number of MRI-guided and ultrasound-guided biopsy should be shown in the manuscript.
Done. The sentence “Imaging guided target prostate biopsy included in-bore MRI guided prostate biopsy (MRGB) (January 2012 to August 2014) and Ultrasound/MRI (US/MRI) fusion guided prostate biopsy (September 2014 to present).” has been changed to “Imaging guided target prostate biopsy included in-bore MRI guided prostate biopsy (MRGB) (January 2012 to August 2014, n=186) and Ultrasound/MRI (US/MRI) fusion guided prostate biopsy (September 2014 to September 2018, n=446).” In Line 123 – 124.
- What are the purpose of showing Figure 3, 4, and 5? Are these representatives? There are no methods or results descriptions of the Figures in the result section of the manuscript according to the Figures. If it is representatives of some special methods or results, the authors should describe them according to the Figures. If it is just the representatives of the MRI-guided and ultrasound-guided biopsy, are these Figures necessary? Because there is no novel information in these Figures.
Done. Figure 3, 4, and 5 are examples of US/MRI fusion biopsy and MRI-guided biopsy. These cases have proven the concept of our study that TRUS prostate biopsy would miss the lesions in the anterior and inferior aspect of prostate, but the target biopsy showed no problem to sample them (Figure 3, 4, 5), even the lesion could be very small (<1 cm, Figure 4). In addition, the manuscript is submitted for the special issue of Diagnostic Biomarkers in Prostate Cancer 2020, which may attract a lot of radiologists to read who love images. Nevertheless, if the editor thinks the figures are not necessary, we will accept the editor’s decision. Thanks.
No change made.
- There is no Table showing the results of logistic regression analysis. Please add and explain it.
Done. A table showing the results of logistic regression analysis with explanation was added as table 2 in Line 214.
Table 2. Logistic regression of upgrading probability on PI-RADS score.
|
|
Coef estimate |
Std error |
P value |
|
Intercept |
-8.004 |
2.54 |
0.0016 |
|
PI-RADS |
2.245 |
0.64 |
0.00054 |
The sentence “In the range of PI-RADS score 3 to 5, each unit increase in PI RADS score is predictive of 2.25 unit increase in log odds of upgrading.” has been changed to “As logistic regression model uses a linear relationship between the log odds and the score, in the range of PI-RADS score 3 to 5, each unit increase in PI RADS score is predictive of 2.25 unit increase in log odds of upgrading (table 2).” in Line 211 – 213.
Minor revisions;
- In Figure 1, an abbreviation of bx should be spelled out.
Done. “..bx..” changed to “…biopsy..” in Figure 1.
- In Figure 1, the percent of 354 patients should be shown in N = 37, 198, 11, 31, 27, and 4 except for Upgrading patients of N = 77.
Done. The percent of 354 patients “N = 37 (10%), 198 (56%), 119 (34%), 11 (11/119, 9%) have been added to the table. In order to avoid confusion, we believed that no need to add percentage for the last 3 numbers (31, 27 and 4) in the table.
Thanks.
Round 2
Reviewer 4 Report
Comments for Manuscript ID:
diagnostics-824832
The authors addressed the reviewer’s previous comments. However, there are still several points to correct.
- In Figure 1, N = 77 (22%) should be N = 77 (77/108, 71%) or all the percentage should be based on N = 354.
- In Figure 3, 4, and 5, PI-RAZ scores should be shown.
- In line 122, 182 MRGB and 446 US/MRI biopsies were performed in your study group, however, how many in 108 patients with abnormal MRI (N = 119) in this study?
Author Response
Reviewer 4:
The authors addressed the reviewer’s previous comments. However, there are still several points to correct.
1. In Figure 1, N = 77 (22%) should be N = 77 (77/108, 71%) or all the percentage should be based on N = 354.
Done. In order to avoid confusion, N=77 (22%) has been changed to N=77 (77/354, 22%) in Figure 1.
2. In Figure 3, 4, and 5, PI-RAZ scores should be shown.
Done. In Figure 3, (PI-RADS 5) was added in Line 182.
In Figure 4, (PI-RADS 4) was added in Line 194.
In Figure 5, (PI-RADS 5) was added in Line 202.
3. In line 122, 182 MRGB and 446 US/MRI biopsies were performed in your study group, however, how many in 108 patients with abnormal MRI (N = 119) in this study?
Done. “In line 122, 182 MRGB and 446 US/MRI biopsies were performed in your study group” indicated during that time period, in my institute, we did these target imaging guided biopsies. Some of them were not in AS, so not in my study group.
All 108 patients had CSRs at MRI, so they underwent imaging guided prostate biopsies. Please see the sentences in Line 25 and 26. Thanks.
No change made.